# Different SO$_2$ Doses and the Impact on Amino Acid and Volatile Profiles of White Wines

**Cátia V. Almeida Santos [1], Catarina Pereira [2] , Nuno Martins [3] , Maria João Cabrita [3],\* and Marco Gomes da Silva [1],\***

1   LAQV-REQUIMTE, Departamento de Química, Faculdade de Ciências e Tecnologia, Universidade Nova de Lisboa, Campus de Caparica, 2829-516 Caparica, Portugal; cv.santos@campus.fct.unl.pt
2   MED—Mediterranean Institute for Agriculture, Environment and Development, Instituto de Investigação e Formação Avançada, Universidade de Évora, Pólo da Mitra, Ap. 94, 7006-554 Évora, Portugal; ccd.pereira@campus.fct.unl.pt
3   MED—Mediterranean Institute for Agriculture, Environment and Development & CHANGE—Global Change and Sustainability Institute, Departamento de Fitotecnia, Escola de Ciências e Tecnologia, Universidade de Évora, Pólo da Mitra, Ap. 94, 7006-554 Évora, Portugal; nmartins@uevora.pt
\*   Correspondence: mjbc@uevora.pt (M.J.C.); mdr@fct.unl.pt (M.G.d.S.)

**Abstract:** SO$_2$ is a preservative often used in the food industry, particularly in the wine industry. However, regulatory authorities and consumers have been strongly suggesting its reduction or even its replacement. In order to understand the impact of SO$_2$ on the profiles of volatile organic compounds (VOCs) as well as amino acids (AAs), the aging of two white wines (one being a varietal and another being a blend) under identical conditions and in the presence of different doses of total SO$_2$ was studied. After alcoholic fermentation (t = 0), either no SO$_2$ was added (0 mg/L), or 30, 60, 90, or 120 mg/L of SO$_2$ was applied. The samples under study were kept on fine lees for 3 months (t = 3). After 3 months (t = 6) and 9 months (t = 12), the wines were bottled and analyzed. For t = 0 and t = 3, the samples were submitted to HS-SPME-GC/MS for VOC analysis and LC-DAD for AA analysis. From the principal component analysis of the detected VOCs, it was observed that the blended wine in comparison with the varietal wine, was less impacted by the applied SO$_2$ doses and aging time. From the AA profile, it was also observed in this study that maturation on fine lees resulted in an increase in the total concentration of AAs as would be expected.

**Keywords:** white wine; sulfur dioxide (SO$_2$); HS-SPME-GC/MS; volatile organic compound (VOC); amino acids (AAs); HPLC-DAD

## 1. Introduction

In Europe, particularly in Mediterranean countries, the wine industry presents considerable economic relevance. In Portugal, production has grown in recent years, either for domestic consumption or exportation. Alentejo is one of the most important wine-growing regions in Portugal, both in terms of area and economic value [1,2].

The recognition of the impact of human activity on the environment has generated the need for more sustainable processes and methodologies. The wine industry, taking into account its specificities, is not an exception [3]. Moreover, there is a trend in the wine sector for consumers to search for wine produced with organic grapes and without the addition of sulfites and/or other additives [4]. Regardless of the misunderstanding and confusion of adjectives such as "organic" and "biological", the advantage is that it is raising awareness of sustainability concepts in wine production, despite its multidimensionality.

In the winemaking process, several oenological products can be used, presenting an impact on CO$_2$ emissions and, therefore, on the carbon footprint of wines [5,6]. One of these products is sulfur dioxide (SO$_2$). Although this additive has been used since the 19th century at several steps during the winemaking process, the need to reduce or replace it

is now emerging [7,8]. In addition to environmental concerns, the excessive exposure of consumers to sulfites through food products increases the report of allergic episodes, and, therefore, overexposure to sulfites increases public health concerns [9].

The European Union, in order to meet health as well as environmental concerns, introduced sulfites in the list of substances and/or products causing allergies or intolerances whenever concentrations are present in values higher than 10 mg/kg or 10 mg/L regarding the total $SO_2$ content [10]. Currently, there is already an indication to label food products when $SO_2$ contents are equal to or greater than 10 mg/kg or 10 mg/L and concomitantly to reduce or replace this additive [11,12]. Replacements for $SO_2$ are being pursued while maintaining the quality of the final product. However, it appears to be difficult to find a compound as versatile as $SO_2$ [13–15].

$SO_2$, among other known characteristics, acts in the winemaking process as an antioxidant and antiseptic [16]. $SO_2$ at wine pH in an acid-base equilibrium system is present in three forms: molecular $SO_2$, bisulfite ($HSO_3^-$), and sulfite ions ($SO_3^{2-}$) [17,18]. The way these three forms act and impact the winemaking process is known and has been extensively described [13,19–22].

The application of different doses of $SO_2$ on the alcoholic fermentation step of the winemaking process, the simultaneous impact on the yeast nitrogen metabolism, and the role of $SO_2$ as a protective agent for yeast against stress conditions during fermentation, such as the increasing presence of ethanol levels, have also been reported [23,24].

Aging and storage conditions influence the amino acid (AA) content of wines. Indeed, keeping wines over lees tends to increase nitrogenated substances, namely AAs, since yeast autolysis occurs and intracellular enzymes slowly hydrolyze yeasts cells, releasing AAs into the wine medium [25,26]. Additionally, when wines undergo cold stabilization, aiming to achieve tartaric stabilization, followed by filtration before bottling, the AA content is reduced [20]. During storage, the VOC profile of wine also changes, affecting wine aroma and quality [27]. $SO_2$ has an important role in preventing microbiological spoilage and oxidation, which both negatively affect the VOC profile and, hence, the organoleptic characteristics of the final product [28].

The main aim of this work was to understand the influence of different doses of $SO_2$ (added after alcoholic fermentation) on VOCs of wine over time, as well as the AA content of wines after 3 months over fine lees. The goal was to evaluate the behavior of wines impacted by $SO_2$ doses in order to be able to safely decrease their addition without compromising quality. Headspace solid-phase microextraction gas chromatography/mass spectrometry (HS-SPME-GC/MS) was used to identify and semi-quantify VOCs. High-performance liquid chromatography–diode array detection (HPLC–DAD) was used for AA detection and quantification.

Two white wines produced in the region of the Alentejo were used, one varietal (Antão Vaz; AV) and one Blend (BL) of Portuguese varieties (Síria (32%), Rabo de Ovelha (17%) and Antão Vaz (10%) were the main varieties presented). The wines were analyzed after 3 months over fine lees (t = 3), then bottled and analyzed again after 3 months (t = 6) and 9 months (t = 12). In order to address the main goal of elucidating the impact of $SO_2$ in wine aging, the results obtained with these wines were compared with those observed for initial wines (t = 0).

## 2. Materials and Methods

### 2.1. Reagents

HPLC-grade acetonitrile and methanol were purchased from VWR International (USA) and from Labscan (Dublin, Ireland), respectively. Aspartic acid (Asp), glutamine (Gln), asparagine (Asn), glutamic acid (Glu), glycine (Gly), threonine (Thr), arginine (Arg), alanine (Ala), γ-aminobutyric acid (GABA), proline (Pro), tyrosine (Tyr), valine (Val), methionine (Met), cysteine (Cys), leucine (Leu), phenylalanine (Phe), ornithine (Orn), lysine (Lys), internal standard (IS) (L-2-aminoadipic acid), and derivatizing agent diethyl ethoxymethylenemalonate (DEEMM) were analytical grade and purchased from Sigma

Aldrich (St. Louis, MO, USA). Sodium hydroxide was purchased from VWR International (Radnor, PA, USA), sodium azide and boric acid were purchased from Sigma Aldrich (St. Louis, MO, USA), glacial acetic acid (analytical grade) was purchased from Fisher Scientific (Porto Salvo, Portugal), and hydrochloric acid was purchase from Honeywell, Fluka (Morris Plains, NJ, USA). Ultrapure water was generated by a Milli-Q system Millipore (Bedford, MA, USA). All solutions were filtered through 0.45 μm nylon membranes filters (Whatman, Merck, Darmstadt, Germany) and degassed before use.

### 2.2. Wine Samples

Two white wines from 2018, one varietal (Antão Vaz; AV) and one Blend (BL- Síria (32%), Rabo de Ovelha (17%), Antão Vaz (10%), and Viosinho, Fernão Pires, Arinto, Verdelho, Manteúdo, Gouveio, Semillon, Perrum and Diagalves, all under 6%) were used in this study. All these varieties are allowed for the production of wines under the "Alentejo" Denomination Origin [2]. These wines were chosen since it is common in the industry to mix grape varieties from the same region in the fermentation process for commercial wines, as well as to produce varietal wines. Wines were produced by CARMIM—Cooperativa Agrícola de Reguengos de Monsaraz, Portugal. Both wines were fermented in large volumes (500,000 L) following CARMIM vinification protocols, and the scale of fermentation was chosen to mimic what would happen in an industrial environment. Fermentation of Antão Vaz occurred with the addition of yeast Alchemy II (25 g/hL), and for Blend wine, fermentation took place with the addition of KB12 yeast (25 g/hL). Both wines fermented at 18 °C under similar conditions.

At the end of alcoholic fermentation, when reducing sugars were less than 2 g/L, a portion of the wine was separated and distributed among several glass carboys, and $SO_2$ at different doses (0, 30, 60, 90, and 120 mg/L) was applied using a commercial 6% solution (SAI, SOLFOX 6 N° CE: 231-870-1). Total and free $SO_2$ analyses were performed by iodometric titration (method OIV-MA-AS323-04B) according to OIV methods [29]. This is a simple and fast method, widely used in cellars for the analysis of wines. Wines were kept in contact with fine lees (at 16 °C) for 3 months. After this period, wines were bottled and stored at 16 °C for 3 and 9 months (the aging periods in the bottle of this study). The VOCs and AAs were analyzed at the end of alcoholic fermentation (t = 0) and after 3 months over fine lees (t = 3). VOCs were also analyzed in wines after 3 months (t = 6) and 9 months (t = 12) of aging in the bottle. Three bottles of each wine were used at each sampling time and analyzed in duplicate. Sampling times followed a common period of time over lees and were extended for one year.

### 2.3. HS-SPME Sampling of Wine Volatiles

HS-SPME sampling experiments were carried out, as reported elsewhere [15]. Briefly, a 1 cm DVB/CAR/PDMS SPME fiber, 50/30 μm film thickness ($d_f$), supplied from Supelco (Bellefonte, PA, USA) was used for VOCs extraction by exposing it to the headspace of the sample. Before use, the fiber was conditioned following the manufacturer's recommendations. Fiber blanks were performed periodically to ensure that contaminants and/or carryovers were absent. A total of 2 g of sodium chloride was added to 5.0 mL of the sample in a 22 mL vial and sealed with a Teflon-lined rubber septum/magnetic screw cap. The vial was equilibrated for 5 min at 30 °C and then extracted for 30 min at the same temperature. Thermal desorption of the analytes was carried out by exposing the fiber in the GC injection port at 260 °C for 3 min in a splitless mode for the same time. All samples were analyzed in duplicate.

### 2.4. GC/MS Analysis

The analyses were performed on a GC/MS system consisting of a Bruker GC 456 with a Bruker mass selective detector Scion TQ as reported in a previous work elsewhere [15]. A CTC Analysis autosampler CombiPAL was used. Data were acquired with MSWS 8.2 Bruker and analyzed with Bruker MS Data Review 8.0. Chromatographic separation was

achieved on a SupelcoWaxTM 10 PLUS capillary column (60 m × 0.25 mm i.d., 1.0 μm $d_f$) supplied by Supelco Analytical (Supelco, Bellefonte, PA, USA). The oven temperature program began at 40 °C, held for 5 min, raised at 4 °C/min up to 240 °C, and held for 5 min. Helium was used as carrier gas at a constant flow of 1.7 mL/min at the electronic flow control (EFC 21). The MS transfer line and source temperatures were set at 260 °C. Spectra were matched with NIST MS Search Program Version 2.3. To determine the retention times and characteristic mass fragments ($m/z$), electron ionization (EI) at 70 eV was used, and mass spectra of the analytes were recorded at full scan, in the range of 40 to 450 Da. The linear retention index values (LRIs) were calculated through analysis of the commercial hydrocarbon mixture (C8–C20) under the same chromatographic conditions. The relative amounts of individual components are expressed as percent-peak areas relative to the total peak area of the chromatogram (Relative Peak Area—RPA). Only compounds presenting a relative area above 0.0002% were considered for statistical treatment.

### 2.5. Amino Acids Analysis by HPLC-DAD

Samples were derivatized as described in previous works [30,31]. The aminoenone derivatives were analyzed using HPLC–DAD equipment (Waters Alliance System 2695 series equipped with a Separation Module with an Alliance Series Column Heater and a Photodiode Array Detector—2998 PDA Detector from Waters, Milford, MA, USA). The analytical column used was an ACE HPLC C18 column (4.6 × 250 mm, 5 μm particle size). After separation and detection, the AAs peaks were quantified, considering the respective peak areas and calibration curves, as described previously [31]. The validation of the chromatographic method is also described in Pereira et al. 2021 [31]. The calibration curves used for each AA present determination coefficients ($R^2$) from 0.994 to 0.998, as described previously [31], with some modifications on the ranges of the calibration curves.

### 2.6. Statistical Analysis

Principal components analysis (PCA) was performed using XLSTAT Version 2020.5.1 by Addinsoft to reduce the number of variables (percentage of the peak areas) to detect a pattern in the relationship between the variables (compounds) and the wine samples (which are the objects of the study).

One-way analysis of variance (ANOVA), using the multiple comparison Bonferroni test, was performed to compare the means at the level of significance of $p < 0.05$ for total and free $SO_2$ content evolution in the wine samples under study. Two-way analysis of variance (ANOVA), using the multiple comparison Bonferroni test, was performed to compare the means at the level of significance of $p < 0.05$ for VOCs and AAs evolution. For ANOVA analysis, two factors were considered: $SO_2$ doses (0, 30, 60, and 120 mg/L) and time (0, 3, 6, and 12 months after fermentation) when applicable. GraphPad Prism version 9.0.0 by GraphPad Software was used for this purpose.

## 3. Results and Discussion

### 3.1. Volatile Organic Compound

3.1.1. $SO_2$ in Wine Samples

After alcoholic fermentation, for the two wines under study, monovarietal wine Antão Vaz (AV) and blend wine (BL), total and free $SO_2$ content were analyzed. Before further $SO_2$ addition (t = 0), total and free $SO_2$ content was, for AV, 32 and 18 mg/L and for BL, 12 and 6 mg/L—total and free $SO_2$ content, respectively. After adding $SO_2$ at different doses (0, 30, 60, 90, and 120 mg/L of $SO_2$), the evolution of these parameters was followed using the same iodometric titration method for 3, 6, and 12 months after fermentation. Figure 1 summarizes the $SO_2$ content (Table S1 on Supporting Material) of both wines over time. It is possible to observe that both $SO_2$ forms (total and free) decrease with time in the wine samples (Blend and varietal wines). The decrease in total

SO$_2$ can possibly be related to its effectiveness as an antioxidant over time, which is in accordance with other authors [20,32].

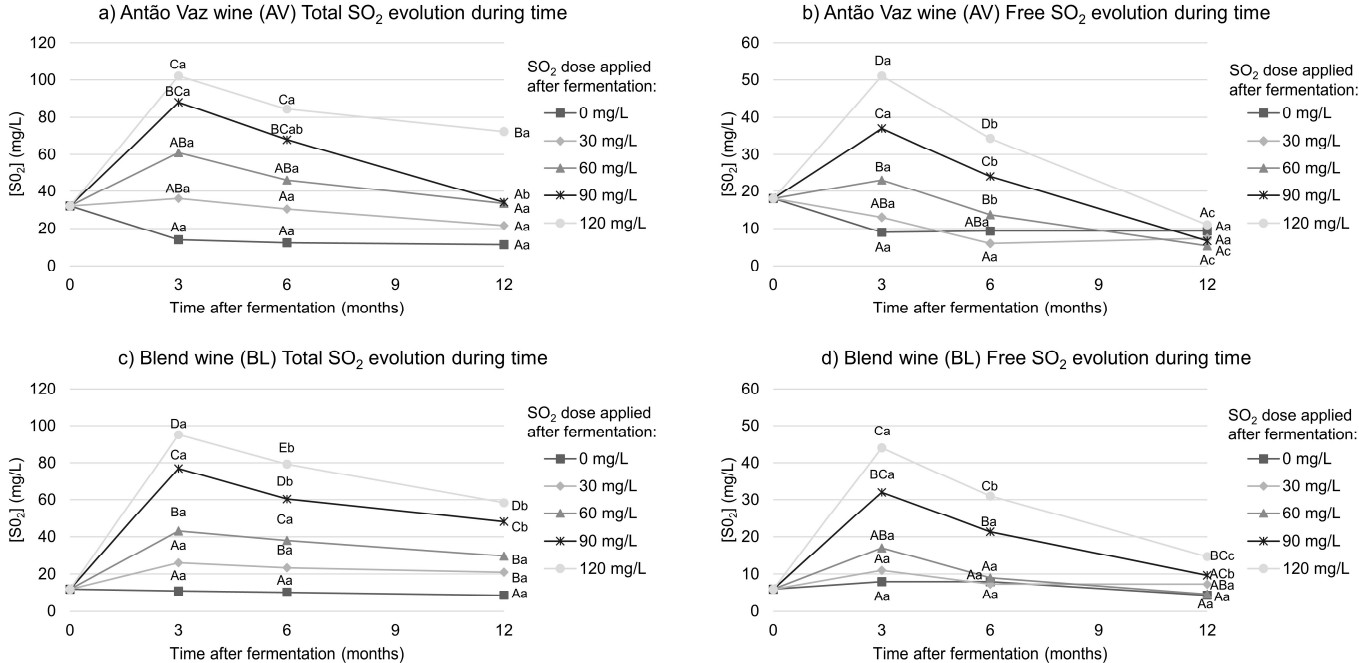

**Figure 1.** SO$_2$ total and free concentration evolution during time of aging. On (**a**,**b**) are represented the Antão Vaz wine (AV), total and free forms, respectively. On (**c**,**d**) are represented the Blend wine (BL), total and free forms, respectively. The statistical results are presented on the graphic: different capital letters (A,B,C,D,E) means that each value differs significantly (one-way variance analysis, $p < 0.05$) according to SO$_2$ dose for each time, and different lowercase letters (a–c) means that each value differs significantly (one-way variance analysis, $p < 0.05$), according to time after fermentation for each SO$_2$ doses.

When performing ANOVA on the average data for total and free SO$_2$ on each wine, some significant differences were observed considering the aging time and the different SO$_2$ doses applied. Both total and free SO$_2$ for AV and BL samples showed a decrease in their content over time, no matter the added SO$_2$ dose applied, as already described elsewhere with different wine samples and varieties [33].

### 3.1.2. Analysis of Volatile Organic Compounds

The HS-SPME-GC/MS method was applied to study the VOCs profile of wine samples and the influence of SO$_2$ doses and aging time. VOCs were tentatively identified by matching mass spectra with spectra of reference compounds in the NIST library. LRIs were also calculated using a commercial hydrocarbon mixture (C8–C20) and compared with LRIs described in the literature [15,34–39]. Table 1 shows the information regarding VOCs, by compound number, tentatively identified in the analyzed samples of AV and BL if they have appeared, at least, in one sample under study.

### 3.1.3. Volatile Organic Compounds in Wines

In this work for AV and BL wines, the relative amount of each analyte was calculated as the percentage ratio of the respective peak area in relation to the total peak area (RPA) of the chromatogram (Table S2 on Supporting Material). Only compounds that present an RPA above 0.0002% were considered for further statistical treatment. It was possible to tentatively identify 83 VOCs that were present in at least one sample. The analyses of the VOCs profile for AV and BL wines indicate that esters (47 compounds), alcohols (14 compounds), carboxylic acids (4 compounds), ethers (2 compounds), ketones

(2 compounds), and aldehydes (2 compounds) were the most tentatively identified chemical groups, together with 12 miscellaneous compounds. Additional 51 compounds (unknowns) were also detected, but their identification assignments could not be carried out since the library match was below 850. The limit of 850 was the criterion established as the minimum limit to accept the tentative identification obtained from the library of spectra, together with the calculated LRI values.

**Table 1.** VOCs tentatively identified in all analyzed samples of AV and BL wines that were found in at least one sample.

| Compound n° | LRIcal [LRIlit] [a] | Compound [Common Name] | Most Abundant Ions (*m/z*) |
|---|---|---|---|
| | | Esters | |
| 1 | 892 [863–893] | Ethyl Acetate | 43/61/70 |
| 2 | 957 [955–975] | Ethyl 2-methylpropanoate [Ethyl isobutyrate] | 45/57 |
| 3 | 971 [924–985] | Propyl acetate | 43/61/71 |
| 4 | 1008 [963–1018] | 2-Methylpropyl acetate | 43/56/71 |
| 5 | 1031 [978–1045] | Ethyl butyrate | 71/43/88/73/41 |
| 6 | 1057 [1009–1066] | Ethyl 3-methylbutyrate [Ethyl isovalerate] | 57/41/70/88 |
| 7 | 1062 [1013–1071] | Butyl acetate | 43/56/61 |
| 8 | 1121 [1071–1131] | 3-Methylbutyl acetate [Isoamyl acetate] | 43/55/70 |
| 9 | 1140 [1125–1176] | Pentyl acetate [Amyl acetate] [c] | 43/55/70 |
| 10 | 1177 [b] | Methyl 4-methylvalerate [d] | 57/74/43/87 |
| 11 | 1189 [1143–1175] | Methyl hexanoate [c] | 74/43/87/55 |
| 12 | 1234 [1198–1244] | Ethyl hexanoate | 88/99/43/70/60 |
| 13 | 1268 [1251–1287] | Hexyl acetate | 43/56/69/61 |
| 14 | 1278 [b] | 3-Methylbutyl butyrate [Isoamyl butyrate] [d] | 71/43/105/55 |
| 15 | 1287 [b] | Ethyl 4-hexenoate isomer | 68/55/41 |
| 16 | 1295 [1283–1305] | Ethyl 3-hexenoate isomer | 41/69/55 |
| 17 | 1301 [b] | Acetate 4-hexenoate isomer | 67/82/43 |
| 18 | 1312 [1292–1307] | Acetate 3-hexenoate isomer | 67/43/82 |
| 19 | 1329 [1304–1322] | Ethyl heptanoate [Grape oil] | 88/43/70/113/101/60 |
| 20 | 1346 [1327–1353] | Ethyl 2-hexenoate isomer | 55/99/73/41 |
| 21 | 1360 [1304–1322] | Heptyl acetate [d] | 43/70/56 |
| 22 | 1386 [1351–1391] | Methyl octanoate | 74/87/43/55 |
| 23 | 1430 [1402–1454] | Ethyl octanoate | 88/57/43/127 |
| 24 | 1455 [1455–1472] | 3-Methylbutyl hexanoate [Isoamyl hexanoate] | 70/43/99/55 |
| 25 | 1460 [1429–1489] | Octyl acetate | 43/56/70/83 |
| 26 | 1515 [1508–1538] | Propyl octanoate | 61/145/127/41 |
| 27 | 1531 [1511–1561] | Ethyl nonanoate [Wine ether] | 88/101/70/41/55 |
| 28 | 1534 [1525–1576] | 2-Methylpropyl octanoate [Isobutyl octanoate] | 56/41/127/145 |
| 29 | 1555 [1547–1560] | Octyl formate [c] | 41/55/69/83 |
| 30 | 1565 [1583–1594] | Hexyl octanoate [d] | 41/56/127/69/145 |
| 31 | 1591 [1570–1636] | Methyl decanoate | 74/87/43/55 |
| 32 | 1631 [1636–1680] | Ethyl decanoate | 88/70/55 |
| 33 | 1643 [1660–1693] | 3,7-Dimethyl-6-octen-1-yl acetate [Citronellol acetate] [c] | 41/69/81/55/95 |
| 34 | 1654 [1657–1695] | 3-Methylbutyl octanoate [Isoamyl caprylate] | 70/127/43 |
| 35 | 1661 [1642–1691] | Decyl acetate [c] | 43/69/55/83 |
| 36 | 1667 [1622–1680] | Diethyl succinate | 101/129/55 |
| 37 | 1686 [1663–1727] | Ethyl 9-decenoate isomer | 55/88/135 |
| 38 | 1731 [1724–1747] | Propyl decanoate | 61/173/41/155 |
| 39 | 1754 [1637–1772] | Ethyl undecanoate [d] | 41/88/55/70/101 |
| 40 | 1815 [1804–1833] | Methyl dodecanoate [Methyl laurate] | 74/43/127 |
| 41 | 1827 [1782–1852] | 2-Phenylethyl acetate | 104/91/43 |
| 42 | 1838 [1837–1881] | Ethyl dodecanoate [Ethyl laurate] | 88/70/41 |
| 43 | 1859 [1840–1897] | 3-methylbutyl decanoate [Isoamyl decanoate] | 70/43/55 |
| 44 | 1908 [1841] | Ethyl isopentyl succinate | 101/129/55 |
| 45 | 1953 [b] | Ethyl tridecanoate [c] | 60/73/88 |
| 46 | NC [2057–2062] | Pentyl laurate [Amyl laurate] [c] | 43/70/55/143 |
| 47 | NC [2233–2242] | Ethyl hexadecanoate [Ethyl palmitate] | 88/41/157 |
| 48 | NC [2241–2274] | Ethyl tetradecanoate [Ethyl myristate] [d] | 41/88/70/157 |
| | | Ethers | |
| 49 | 1090 [b] | 1-(1-Ethoxyethoxy)pentane [Acetaldehyde ethyl amyl acetal] | 73/45 |
| 50 | 1756 [b] | Octyl ether | 57/71/41/83 |
| | | Ketones | |
| 51 | 991 [987–991] | Butanedione [c] | 43/61/86 |
| 52 | 1384 [1386–1387] | 2-Nonanone [d] | 58/43 |

**Table 1.** *Cont.*

| Compound n° | LRIcal [LRIlit] [a] | Compound [Common Name] | Most Abundant Ions (*m/z*) |
|---|---|---|---|
| | | Alcohols | |
| 53 | 930 [927–968] | 2-Propanol | 45/44/43 |
| 54 | 1023 [1012–1032] | 2-Butanol [d] | 45/59/43 |
| 55 | 1066 [1047–1111] | 2-Methylpropyl alcohol [Isobutanol] | 41/55/73 |
| 56 | 1128 [1102–1175] | 1-Butanol | 43/56/70 |
| 57 | 1197 [1173–1211] | 3-Methylbutan-1-ol [Isopentyl alcohol] | 55/41/70 |
| 58 | 1307 [1313–1357] | 3-Methylpentan-1-ol | 56/96/41 |
| 59 | 1340 [1292–1348] | 1-Hexanol | 56/41/69 |
| 60 | 1375 [1358–1379] | 3-Hexen-1-ol | 67/41/82/55 |
| 61 | 1433 [1428–1457] | 1-Heptanol [d] | 70/55/41/88 |
| 62 | 1635 [1630–1694] | 1-Nonanol [d] | 55/41/70/83/97 |
| 63 | 1736 [1720–1794] | 1-Decanol | 41/55/69/83 |
| 64 | 1850 [1760–1799] | 3,7-Dimethyloct-6-en-1-ol [Citronellol] [d] | 41/67/55/81/95 |
| 65 | 1889 [1846–1870] | Phenylmethyl alcohol | 108/79 |
| 66 | 1918 [1873–1947] | Phenethyl alcohol | 91/65/122 |
| | | Aldehydes | |
| 67 | NC [700–744] | Acetaldehyde [Ethanal] | 44/43 |
| 68 | 1407 [1388–1415] | Nonanal [c] | 41/57/70/82 |
| | | Carboxylic acids | |
| 69 | 1918 [1935–1965] | Heptanoic acid | 60/73/41 |
| 70 | NC [2051–2091] | Octanoic acid | 60/70/41/101 |
| 71 | NC [2269–2276] | Decanoic acid | 73/41/129 |
| | | Miscellaneous | |
| 72 | 1335 [1309–1363] | Ethyl 2-hydroxypropanoate [Ethyl lactate] | 45/75 |
| 73 | 1366 [1369–1409] | 3-Ethoxypropan-1-ol | 58/45/71 |
| 74 | 1489 [1426–1485] | Furfural | 96/95 |
| 75 | 1541 [1510–1552] | 2-Methylthiolan-3-one [d] | 60/116 |
| 76 | 1556 [1541–1600] | β-Linalool | 93/71/55/41/121 |
| 77 | 1635 [1618–1621] | Ethyl 2-furylcarboxylate | 95/112 |
| 78 | 1711 [1704–1715] | Terpineol isomer | 59/93/121/136 |
| 79 | 1718 [1698–1755] | 3-(Methylsulfanyl)-1-propanol [Methionol] | 106/57/45/73 |
| 80 | 1783 [1741–1778] | 1,2-Dihydro-1,1,6-trimethylnaphthalene [TDN] [c] | 157/142 |
| 81 | 1824 [1765–1803] | Methyl 2-hydroxybenzoate [Methyl salicylate] [d] | 120/92/152/65 |
| 82 | 1867 [1816–1833] | β-Damascenone | 69/121/41 |
| 83 | NC [2042–2057] | Nerolidol isomer | 41/69/93/107 |

[a] Linear retention indices calculated from C8–C20 n-linear alkanes; [b] Identification by NIST comparison; [c] Compound observed only on AV wines; [d] Compound observed only on BL wines; LRIlit—Linear Retention indices reported in the literature for WAX capillary column; NC: not calculated since LRI were calculated in the range from C8 to C20 n-linear alkanes [6,15,31,34,37,38,40–51].

The three most representative groups of VOCs are esters, alcohols, and miscellaneous compounds [52,53]. Esters represent, by far, the group with the largest number of compounds, in which the more important contributors are ethyl acetate (8), ethyl hexanoate (12), isoamyl butyrate (14), ethyl octanoate (23) and ethyl decanoate (32). This class of compounds is related to the fermentative process, promoting different fruity aroma characteristics to the wines. This is indeed clear in the case of compounds 12 and 14, which were described to be often present above their sensory threshold [54]. Alcohols were the second more representative chemical class, in which 2-propanol (53), isopentyl alcohol (57), and phenethyl alcohol (66) present the higher percentual content. Indeed, esters, namely, ethyl esters and alcohols, were already reported as the two most important groups building the flavor bouquet of wines. Their presence is also an indicator of the important role of $SO_2$ after the fermentation process by preserving these compounds and inducing the evolution of the floral and fruity notes that they confer on wines [55,56].

Performing an ANOVA analysis of VOCs for each compound in the different aging time periods and $SO_2$ conditions for each wine, it was possible to identify 14 compounds with a statistically relevant difference with a significance of $p < 0.05$. For both wines, the compounds: ethyl acetate (1), isoamyl acetate (8), ethyl hexanoate (12), ethyl octanoate (23), ethyl decanoate (32), 2-propanol (52), isopentyl alcohol (57), phenethyl alcohol (66) and octanoic acid (70) presented at least one statistical difference regarding aging time and/or $SO_2$ conditions. It was also observed that hexyl acetate (13) and ethyl laurate (42) for AV

wines and ethyl isovalerate (6) diethyl succinate (36) and ethyl lactate (72) for BL wines showed statistically relevant differences, although all of them are present in both wine samples. Diethyl succinate (36) on BL wines increased with storage time and decreased when higher doses of $SO_2$ were applied for the same aging period. For compounds ethyl hexanoate (12), hexyl acetate (13), and ethyl decanoate (32), a decrease in both wines during the time of aging was observed, possibly due to their hydrolysis. Compounds ethyl acetate (1) and ethyl lactate (72) increased during the aging time, which was also reported by Cassino et al., 2019 [57]. In the case of ethyl lactate (72), it was observed that not only the aging time but also higher doses of $SO_2$ lead to its higher content.

Regarding the alcohol group family, isopentyl alcohol (57) and phenethyl alcohol (66) increased with aging time, while 2-propanol (53) decreased. Octanoic acid (70) increased during the aging time, possibly by hydrolysis of ethyl octanoate (25) and hydrolysis of octyl acetate (23), and subsequent oxidation of the octanol moiety.

### 3.1.4. Principal Component Analysis of Volatile Organic Compounds during Bottle Ageing

A principal component analysis (PCA) was performed to verify if the different $SO_2$ doses applied during aging could influence the VOCs profile of the wine. In order to build the PCA for each wine group (AV and BL) illustrated in Figures 2 and 3, the relative areas of the tentatively identified VOCs (Table 1) were used. These relative areas were normalized against the relative areas of the initial wine, according to Santos et al., 2022 [34]. Hence, the wine after fermentation (t = 0) is placed in PC1 = 0 and PC2 = 0. It is possible to observe that 74.96% and 66.65% of the system variance was explained by the 1st and 2nd principal components (PC1 and PC2, respectively) for AV wines and for BL wines, respectively.

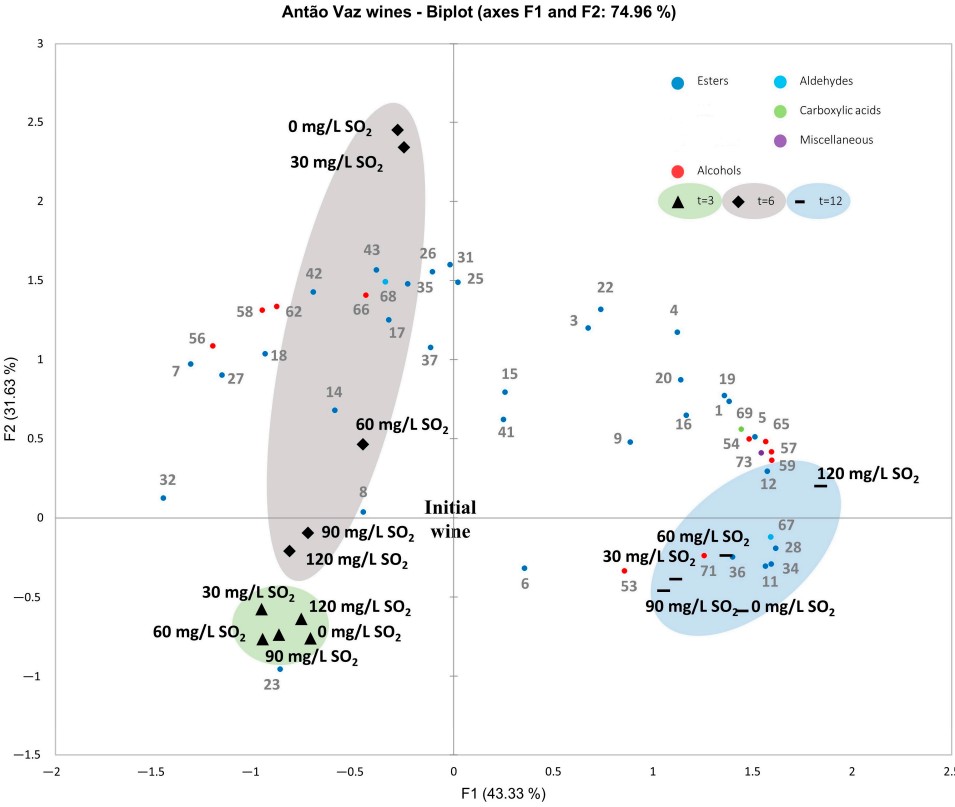

**Figure 2.** Principal component biplot illustrating the simultaneous projection of the wine and volatile organic compounds of Antão Vaz wine (AV). Black dot—initial wine; Black triangle—wines aging for 3 months; Black rhombus—wines aging for 6 months; Black cross—wines aging for 12 months; Dark blue dots—esters; Dark green dots—ethers; Yellow dots—ketones; Red dots—alcohols; Light blue dots—aldehydes; Light grey dots—carboxylic acids; Purple dots—miscellaneous.

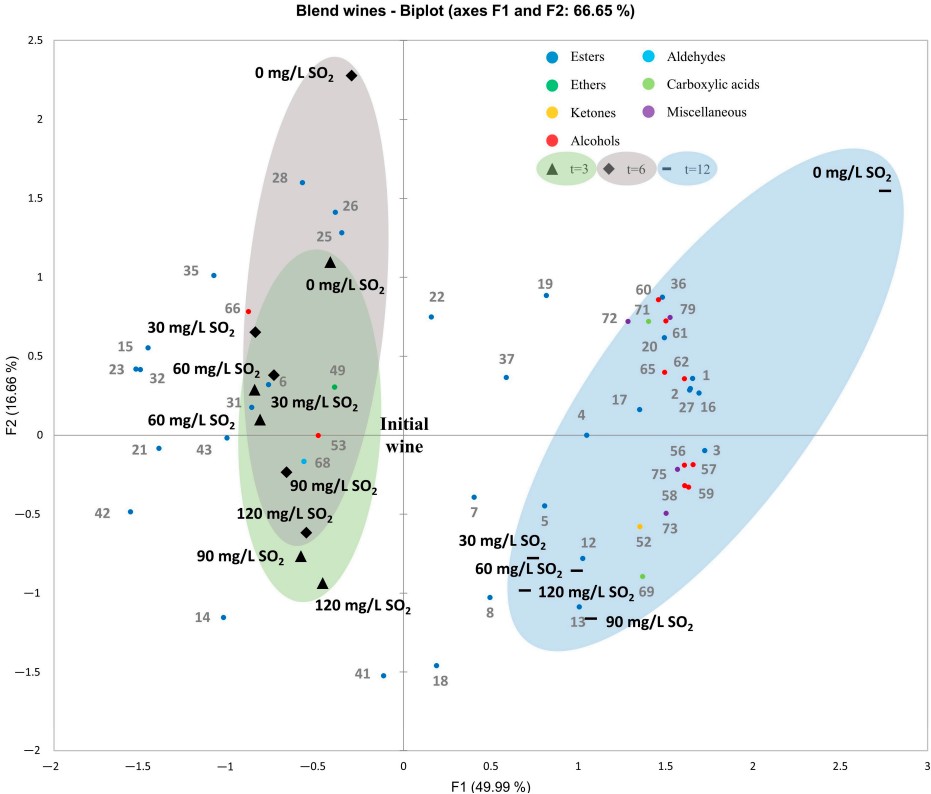

**Figure 3.** Principal component biplot illustrating the simultaneous projection of the wine and volatile organic compounds of Blend wine (BL). Black dot—initial wine; Black triangle—wines aging for 3 months; Black rhombus—wines aging for 6 months; Black cross—wines aging for 12 months; Dark blue dots—esters; Dark green dots—ethers; Yellow dots—ketones; Red dots—alcohols; Light blue dots—aldehydes; Light grey dots—carboxylic acids; Purple dots—miscellaneous.

Figure 2 illustrates the PCA for AV wines. PC1 separates the samples according to the aging time, explaining 43.33% of the system variability, and PC2 separates the samples regarding the application of $SO_2$, explaining 31.63% of the system variability. In the positive quadrant of PC1, samples aged for 12 months are clearly separated from the other wine samples. Along PC2, a fair separation of wines with 3 and 6 months can be observed, being the first ones mainly influenced by ethyl octanoate (23).

However, for samples aged for 6 months, it is possible to observe a distinction of VOCs profile leading to wines well separated according to different $SO_2$ doses applied. The most similar samples to the previous ones (3 months of aging) are the samples with the application of higher doses of $SO_2$ (90 and 120 mg/L), suggesting a protection role of the samples against oxidation, and are characterized by isoamyl acetate (8). For the samples with no addition or addition of 30 mg/L of $SO_2$, samples were similar and mainly influenced by acetate 4-hexenoate (isomer) (17), decyl acetate (35), isoamyl decanoate (43), phenethyl alcohol (66) and nonanal (68). The wines stored with 60 mg/L of $SO_2$ were separated from the others and characterized by isoamyl butyrate (14).

The wines aged for 12 months in the presence of 0, 30, 60, and 90 mg/L of $SO_2$ are very similar. Indeed, they are impacted by diethyl succinate (36), 2-propanol (53), and decanoic acid (71). Samples aged with 120 mg/L are more influenced by ethyl hexanoate (12).

Figure 3 illustrates the PCA obtained for BL wines. PC1 separates the samples according to time of aging, explaining 49.99% of the system variability, and PC2 explaining 16.66% of the system variability, separates the samples according to the addition of $SO_2$. BL wines aged for 12 months are in the positive quadrant of PC1, clearly separated from other wines, and positively influenced by ethyl butyrate (5), isoamyl acetate (8), ethyl hexanoate (12), and isoamyl butyrate (14). However, BL wines without the addition of

SO$_2$ are on the positive side of both PC1 and PC2 and separated from the other samples aged for 12 months.

For wines aged for 3 and 6 months, their separation is not evident according to the time of aging. Wines with the addition of 90 and 120mg/L of SO$_2$ are on the negative side of PC1 and PC2 for both aging times. Isoamyl butyrate (14) is influencing wines aged for 3 months, and 2-propanol (53) and nonanal (68) are impacting wines aged for 6 months.

The wines aged for 3 months with 30 and 60 mg/L of SO$_2$ are influenced by isoamyl decanoate (43). Wines aged for 6 months with 30 mg/L of SO$_2$ are characterized by ethanal (67), and when 60 mg/L of SO$_2$ was used, the wine samples were influenced by ethyl isovalerate (6).

### 3.2. Amino Acids

#### Amino Acids Analysis

The AAs content in AV and BL wines was also evaluated in this work for initial wines and wines after 3 months on lees. An ANOVA analysis was also performed for each AA, and results are presented in Table S3 on Supporting Material. The most represented AAs were proline and γ-aminobutyric acid for both wines, either in the initial wine or 3 months over lees after the addition of different doses of SO$_2$, as already reported by Sartor et al., 2021 [25] for other wine varieties. Regarding ANOVA results, one observed, for both wines, that aspartic acid, glutamic acid, asparagine, γ-aminobutyric acid, proline, and ornithine present at least one statistical difference, either in relation to the initial wine and/or in relation to the different SO$_2$ doses applied. The differences observed, despite being in the same AAs, did not show the same pattern for the two wines under study. Alanine presented a statistical difference only in BL wines.

Figure 4 represents the histogram of total AAs content with wines after fermentation as a reference. For AV (Figure 4A) and BL (Figure 4B) wines, the time over fine lees leads to an increase of AAs concentrations, as already reported in the literature. Indeed, this behavior can be related to yeast autolysis after the decrease of viable cells during maturation over fine lees [25,26,58]. The main known factors that affect yeast autolysis are pH, temperature, ethanol content, and yeast strain. All these factors were kept constant in these wines; hence our results suggest that SO$_2$ doses may also play a role in AA released by yeast autolysis, despite the value found for AV wine with 120 mg/L of SO$_2$. In Table S4 of the Supporting Material, it is observed that the AAs concentration for both wine samples (AV and BL), considering the SO$_2$ amount, present significant differences at $p < 0.05$. Nevertheless, more studies are ongoing to go deeper into this subject in order to clarify it.

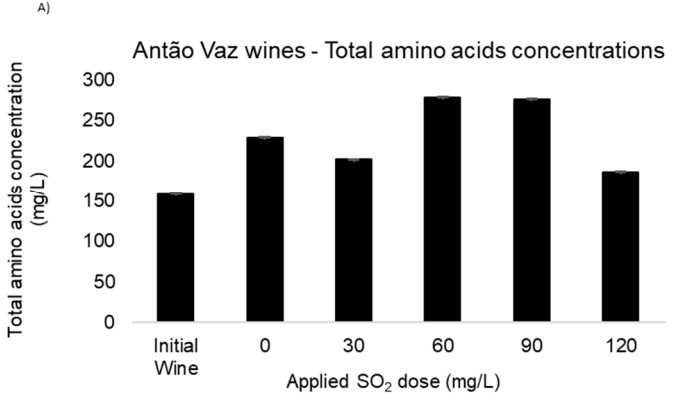
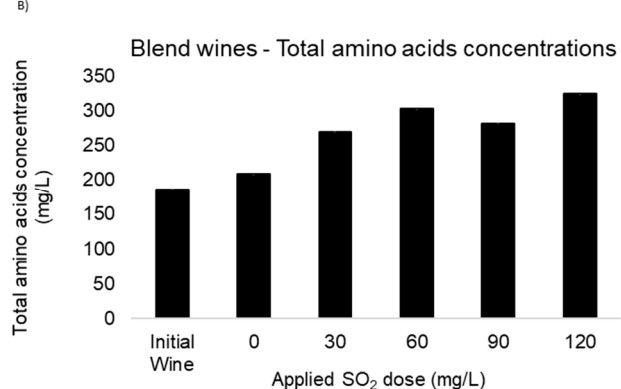

**Figure 4.** Histogram of total amino acids content with wines after fermentation as reference, (**A**) Antão Vaz wines and (**B**) Blend wines. The total concentration was calculated by subtraction and division of total amino acid area for each dose with the total area obtained for wine after fermentation (t = 0) before adding SO$_2$.

Figure 5 illustrates the PCA for (a) AV wines and (b) BL wines with data normalized using wines after fermentation as a reference in order to mitigate the predominance of variables with higher values. The impact of different doses of $SO_2$ is different for the two wine samples. In the case of AV wines (A), PC1 explains 88.43%, and PC2 explains 7.18% of the wine samples distribution. These samples are well spread over the plane defined by the two PCs regarding $SO_2$ addition. Wines with 0 and 30 mg/L of $SO_2$ are on the positive side of PC1 and the negative side of PC2; however, no AA is positively related. The addition of 60 and 90 mg/L of $SO_2$ led to similar wines in the positive side of PC1 and PC2 and was positively influenced by lysine in the case of 60 mg/L of $SO_2$ and proline in the case of 90 mg/L of $SO_2$. The wines with the addition of 120 mg/L of $SO_2$ were more similar to the initial AAs content, although located in a different quadrant.

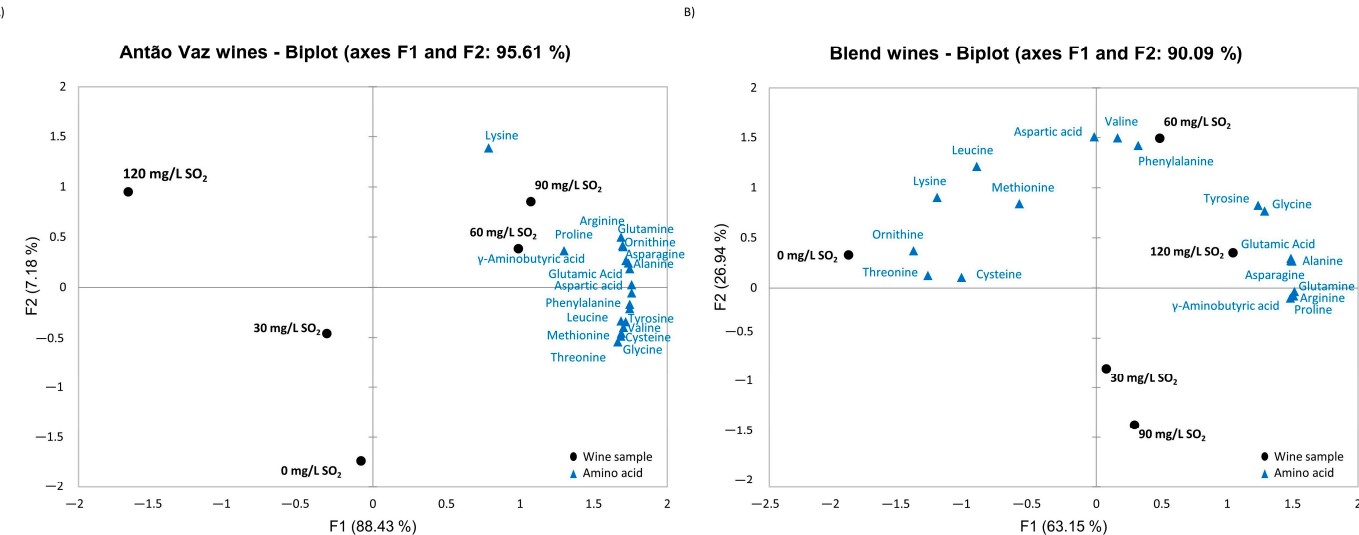

**Figure 5.** Principal component biplot illustrating the simultaneous projection of the wine and amino acids of Antão Vaz (**A**) and Blend wine (**B**). Black dot—wine samples with the respective $SO_2$ concentration applied; Blue triangle—amino acid. Amino acids: Aspartic acid (Asp), glutamine (Gln), asparagine (Asn), glutamic acid (Glu), glycine (Gly), threonine (Thr), arginine (Arg), alanine (Ala), $\gamma$-aminobutyric acid (GABA), proline (Pro), tyrosine (Tyr), valine (Val), methionine (Met), cysteine (Cys), tryptophan (Trp), leucine (Leu), phenylalanine (Phe), ornithine (Orn) and lysine (Lys).

In the case of BL wines (B), PC1 explains 63.15% and PC2 26.94% of the wine samples distribution. In wines without $SO_2$, ornithine and threonine were more affected and presented a higher increase compared to other $SO_2$ condition samples. For 60 mg/L of $SO_2$ addition, aspartic acid, valine, and phenylalanine were positively affected, showing higher values. Wines with 30, 90, and 120 mg/L of $SO_2$ seem to be more similar and positively influenced by proline, arginine, and glutamine. Proline and arginine content increased in a more antioxidant environment, and glutamine content was higher at 90 mg/L of $SO_2$.

The differences observed in the AAs increase among wine samples may be attributed to the different compositions of lees since wines were produced from different musts, which were obtained using different yeasts for the alcoholic fermentation step. Nevertheless, our results reinforce the beneficial effect of keeping wines over lees, namely by the enrichment in amino acid content.

## 4. Conclusions

Two white wines from Alentejo were aged with different $SO_2$ doses applied after fermentation. The samples were analyzed 3, 6, and 12 months after $SO_2$ addition. When an ANOVA was performed to VOCs percentual content for both wines, ethyl acetate (1), isoamyl acetate (8), ethyl hexanoate (12), hexyl acetate (14), ethyl octanoate (23), ethyl decanoate (31), 2-propanol (52), isopentyl alcohol (56), and phenethyl alcohol (64)

presented, at least, one statistical difference regarding the time of aging and/or SO$_2$ conditions. It was also observed that ethyl laurate (41) for AV and ethyl isovalerate (6), diethyl succinate (35), and ethyl lactate (71) for BL, showed statistically significant differences although being present in both wines.

Analyzing the PCAs for each wine, it is possible to verify that the initial wines are completely different from the aged wines, independently of the addition or absence of SO$_2$. The same is observed for wines aged for 12 months. When wines were aged for 3 and 6 months (in the case of AV), it led to well-distinct samples, probably because AV wines are more susceptible to the oxidation-developed aroma in comparison with Blend wines (BL), which appear to be more stable to that action. Regarding the two factors under study, aging time and different doses of SO$_2$, it is clear that time has a major impact on VOCs profile than the addition of different SO$_2$ doses.

Regarding amino acids profile, proline, arginine, and $\gamma$-aminobutyric acid are the most predominant in both studied wines. In general, and as expected, with maturation over lees, the total AAs concentration increased in comparison with initial wines, but SO$_2$ also influenced the increase of the AAs, especially on the AV wines. The cases where 60 and 90 mg/L of SO$_2$ were applied led to the highest AA concentration.

Considering these results, it seems possible to reduce SO$_2$ amounts without compromising the volatile profiles of wines since time plays a more dominant role than SO$_2$ doses. Lower amounts of SO$_2$ do not compromise the normal evolution of volatiles and the normal reduction of free and total SO$_2$ over time. In fact, taking into consideration this work, in the future, a SO$_2$-reducing practice in winemaking could be attempted, owing this strategy does not negatively impact the evolution of wine aromas.

**Supplementary Materials:** The following supporting information can be downloaded at: https://www.mdpi.com/article/10.3390/beverages9020033/s1, Table S1: SO$_2$ total and free concentration evolution during time of aging. On (a) Antão Vaz wines (AV) total and free forms and (b) Blend wines (BL) total and free forms; Table S2: Relative amount of each compound, calculated as the percentage ratio of the respective peak area in relation to relative to the total-peak area (RPA) of the chromatogram; Table S3: Amino acid quantification in AV—(a) and BL—(b) wines. Table S4: Total amino acid concentration in AV and Blend wines.

**Author Contributions:** Conceptualization, M.J.C. and M.G.d.S.; data curation, C.V.A.S. and C.P.; formal analysis C.V.A.S.; funding acquisition, M.J.C. and M.G.d.S.; investigation, C.V.A.S.; methodology, C.V.A.S., C.P. and N.M.; project administration, M.J.C. and M.G.d.S.; resources, M.J.C. and M.G.d.S.; software, C.V.A.S., M.J.C. and M.G.d.S.; supervision, M.G.d.S. and M.J.C.; validation, C.V.A.S.; roles/writing—original draft, C.V.A.S.; writing—review and editing, M.J.C. and M.G.d.S. All authors have read and agreed to the published version of the manuscript.

**Funding:** This work was supported by National Funds through FCT—Foundation for Science and Technology under the Ph.D. Grant [PD/BD/135081/2017] (Cátia Almeida Santos) and Ph.D Grant FCT 2021.07306.BD (Catarina Pereira); Associate Laboratory for Green Chemistry—LAQV, which is financed by national funds from FCT/MCTES [UID/QUI/50006/2020] and the project [UIDB/05183/2020].

**Data Availability Statement:** Not applicable.

**Acknowledgments:** The authors acknowledge Rui Bicho from Laboratório de Enologia Colaço do Rosário, MED, Universidade de Évora for technical assistance. This research was also anchored by the RESOLUTION LAB, an infrastructure at NOVA School of Science and Technology. All individuals included in this section have consented to the acknowledgement.

**Conflicts of Interest:** The authors declare no conflict of interest.

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
