# Peer review of "Different SO2 Doses and the Impact on Amino Acid and Volatile Profiles of White Wines"

_beverages, doi:10.3390/beverages9020033_

Round 1

Reviewer 1 Report

With the aim to reduce SO2 as preservative in the wine industry, this paper studied the impact of SO2 on volatile and amino acid profiles of two white wines (one varietal and one blend) aged under the same conditions. After fermentation, 0, 30, 60, 90 and 120 mg/L of SO2 were applied.

The paper is of interest and falls within the scope of the journal. It is generally well-designed and well-written. However, conclusions should be more emphasized shedding a light on the importance of the results obtained with the research.

English should be improved. I advise Authors to make the paper revised by a mother-English tongue.

Minor:

-        Line 55: correct mmg/kg;

-        Line 123: correct glass carboyls;

-        Lines 191-192: invert 18 with 32 and 6 with 12;

-        Figure 4: what about statistics?

Author Response

Reviewer 1:

With the aim to reduce SO2 as preservative in the wine industry, this paper studied the impact of SO2 on volatile and amino acid profiles of two white wines (one varietal and one blend) aged under the same conditions. After fermentation, 0, 30, 60, 90 and 120 mg/L of SO2 were applied. The paper is of interest and falls within the scope of the journal. It is generally well-designed and well-written. However, conclusions should be more emphasized shedding a light on the importance of the results obtained with the research.

R: The following sentence (in italic) was added in the conclusions in order to clarify this point: “Indeed, taking in consideration this work, wine producers who already have the application of SO2 in their current practice processes, with higher doses for storage protection, may considered as future practice, to reduce final SO2 applications without compromising the evolution of wine aromas.”

English should be improved. I advise Authors to make the paper revised by a mother-English tongue.

R: Text was reviewed in order to correct spelling errors and typos

Minor:

Line 55: correct mmg/kg;

R: It was corrected

Line 123: correct glass carboyls;

R: It was corrected as “glass carboys”, which are glass recipients.

Lines 191-192: invert 18 with 32 and 6 with 12;

R: It was corrected

Figure 4: what about statistics?

R: A table with the results of the ANOVA analysis of the sums obtained was added in supplementary material (Table S4) which is included in the revised text.

Reviewer 2

Abstract: Please, describe what are VOCs and AAs abbreviations.

R: It was corrected

Please, provide information how the derivatization step that was obtain for the aminoenone derivatives.

R: We rewrote this item (2.5 Amino acids analysis by HPLC-DAD) in order to clarify about the derivatization step.

Figure 1. Authors should explain the abbreviations (Ca, BCa, Db......) used in figures a, b, c and d.

R: Explanations were added in the revised version

The explanation about changes of total and free SO2 should be extended. As it can be seen from the results, almost half and more of the free SO2 concentration decreases. Is it possible to predict the concentration of SO2 after 3, 6 or 9 months knowing the initial added content? How can these results help to wine producers? Is it necessary to add SO2 after 3 or 6 months in order to keep wine safe from oxidation and other processes?

R: In this work, no studies were carried out in this sense, although such a study is in our immediate horizon. However, given the low quantities of free SO2, there is an indication of the need for a possible reinforcement of the addition to protect the wine.

What is the difference between "Initial wine" and wine with 0 mg/L SO2? Initial wine shouldn't be the wine with 0 mg/L added SO2 (without added SO2)?

R: In this work, the wine samples received from the cellar that provided them were considered as initial wines (t =0). As for the wines with 0 mg/L of SO2 added, it was a portion of the received wine that was stored during the study period along with the other application conditions. In this way we intended to have the extreme case of unprotected wine with no application versus application of different concentrations.

Reviewer 2 Report

Abstract: Please, describe what are VOCs and AAs abbreviations. 

Please, provide information how the derivatization step that was obtain for the aminoenone derivatives.

 Figure 1. Authors should explain the abbreviations (Ca, BCa, Db......) used in figures a, b, c and d.

The explanation about changes of total and free SO2 should be extended. As it can be seen from the results, almost half and more of the free SO2 concentration decreases. Is it possible to predict the concentration of SO2 after 3, 6 or 9 months knowing the initial added content? How can these results help to wine producers? Is it necessary to add SO2 after 3 or 6 months in order to keep wine safe from oxidation and other processes?

What is the difference between "Initial wine" and wine with 0 mg/L SO2? Initial wine shouldn't be the wine with 0 mg/L added SO2 (without added SO2)?

Author Response

(The authors gave the same response as above.)

Round 2

Reviewer 2 Report

Authors performed required changes in the text.